# Association between Loneliness and Depression among Community-Dwelling Older Women Living Alone in South Korea: The Mediating Effects of Subjective Physical Health, Resilience, and Social Support

**DOI:** 10.3390/ijerph19159246

**Published:** 2022-07-28

**Authors:** Young Mi Lim, Juha Baek, Sungmin Lee, Jung Sug Kim

**Affiliations:** 1Department of Nursing, Yonsei University Wonju College of Medicine, Wonju 26426, Korea; youngmi@yonsei.ac.kr; 2Department of Health Care Policy Research, Korea Institute for Health and Social Affairs, Sejong-si 30146, Korea; 3Landscape Architecture and Urban Planning, College of Architecture, Texas A&M University, College Station, TX 77840, USA; sungminlee@arch.tamu.edu; 4Department of Nursing, Yeoju Institute of Technology, Yeoju 12652, Korea; nureskim@gmail.com

**Keywords:** older adults living alone, physical health, resilience, social support, loneliness, depression

## Abstract

Social isolation and loneliness are the key risk factors for depression in late life. Older adults living alone and socially isolated are at greater risk for physical and mental health. This study aims to examine the mediating effects of subjective physical health, resilience, and social support on the association between loneliness and depression among the elderly female population living alone in South Korea. We included a total of 308 older women aged 60 years or older who live alone in a medium-sized city in South Korea. The survey data was collected using the validated survey instruments between November 2015 and April 2016. A parallel mediation model was performed to investigate whether physical health, resilience, and social support had mediating effects on the association of loneliness with depression. The findings of this study showed that loneliness was directly and indirectly associated with depression through its association with the subjective physical health, resilience, and social support among the older female population living alone. Our results suggest the importance of supporting community-based programs to improve physical and mental health of the elderly people as a way to minimize the level of loneliness and prevent depression.

## 1. Introduction

South Korea is one of the countries with the fastest aging population in the world [1]. The proportion of the elderly population aged 65 years or older among the total population in South Korea has increased over time from 7% in 2000 to 16.4% in 2020, and is estimated to increase to 34.4% by 2040 [2]. In particular, the percentage of the older people living alone among the total Korean population has doubled between 3.8% in 2000 and 7.9% in 2020 [2]. Among the older people in Korea, 19.8% were living alone and the percentage of women (27.4%) was higher than that of men (9.7%) in 2019 [3].

Earlier studies have shown that the elderly people who live alone and have a poor social network were more likely to experience higher levels of social isolation and loneliness than those living with a spouse or family member [4,5,6,7]. These older adults tend to have an increased risk of mental health issues, including depressive symptoms, cognitive dysfunction, and suicidal ideation, compared to those who do not live alone [6,8,9,10]. According to the recent report from the Ministry of Health and Welfare of Korea, 13.5% of the elderly population in Korea experienced depressive symptoms in 2019 and the older adults living alone had higher depressive symptoms compared to those living with a spouse and children (18.7% vs. 10.4%, 16.8%) [3]. Additionally, the effect of living alone and loneliness on mental health may differ by gender. Loneliness and adverse mental health were more likely to be influenced by feelings of living alone among older men, while social network and social engagement significantly mitigated loneliness and depression among older women [11].

Loneliness and depression are major public health issues for elderly individuals, which could worsen their physical health and quality of life, leading to an increased level of suicide [12,13,14,15]. A number of studies have examined the association between loneliness and depression for the older adults. A study reported a significantly positive association between loneliness and depression among the population aged 60 years or older in Turkey [16]. Another study found that depression was significantly related to emotional loneliness, not with social loneliness in the older people aged 60 years or older in the Netherlands [17]. The other study in Taiwan revealed that loneliness and depression had a bidirectional association among the elderly Taiwanese [18]. However, studies about the association targeting the older population living alone are still limited.

Previous research has explored potential mediators of the association between loneliness and depression for elderly people. Physical health appears to be linked to social isolation and mental health. A growing body of research showed that feelings of loneliness accelerates physiological aging [19] and predicts adverse physical health outcomes, including worse sleep and immune stress, in the elderly people [20], which potentially lead to the onset and the persistence of depression [21,22]. Additionally, resilience, an important protective factor for individuals who cope with adversity [23], was found to be a significant contributor to increase quality of life and reduce mental health symptoms for the older people [24]. Resilience was known to act as a protective factor to mitigate negative emotions like loneliness and help maintain well-being among elderly population [25]. Earlier evidence has shown that resilience could have partially mediating effects on the association between loneliness and depression in the older people [24,26,27]. Finally, social support has been regarded as an important factor to address loneliness and depression in older adults [28]. Social support is generally regarded as the existence or network of family, friends, neighbors, or community members that can give emotional, instrumental, appraisal, and informational aid [29]. Several studies have found that social support partially mediated the relationship of loneliness with depression among the elderly in Asian countries [28,30,31]. 

The previous studies have investigated the direct and indirect relationships between loneliness and depression by considering the mediating effects of physical social support, resilience, and physical health, respectively. However, there are few studies understanding the construct relations comprehensively. Most important, the mechanism on how and to what extent loneliness is associated with depression remains unclear among older adults, in particular older women living alone. There are also very few related studies in the context of South Korea with a rapidly aging population. Thus, our study fills these research gaps in the existing literature by (1) investigating the direct influence of loneliness on the depression of older women aged 60 years living alone in South Korea and (2) exploring the mediating effects of subjective physical health, resilience, and social support, comprehensively. This study is based on a framework rooting in a cognitive-behavioral theory that conceptualizes depressive symptoms as resulting from an interaction of events, cognitive processes, and behaviors. Figure 1 presents the conceptual framework for the mediating effects of subjective health, resilience, and social support on the relationship between loneliness and depression based on the existing cognitive-behavioral model.

## 2. Materials and Methods

### 2.1. Study Design and Participants

This study used a cross-sectional research design to examine the mediating effects of subjective physical health, resilience, and social support on the association between loneliness and depression among community-dwelling older women living alone. The study region was a middle-sized city, which is one of the urban–rural linkages in South Korea (called Yeoju). About 31% of the total population (*n* = 112,393) were the elderly, aged 60 years old or above, and the rate of women (53%) was higher than that of men (47%) among the older people in March 2022. Additionally, most of the older adults living alone in the city were female (about 71.4%) [32]. The original inclusion criteria for the study subjects were as follows: (a) aged 60 years and older; (b) living alone at home (i.e., one person household); (c) not diagnosed with dementia; and (d) having no communication problems. Subjects were recruited from a public health center in the city. 

We collected the data between November 2015 and April 2016 from the community-dwelling older adults living alone. In this study, a visiting nurse at the public health center visited houses of elderly people living alone in the city and conducted an interview survey using a structured questionnaire. A voluntary written consent was obtained for participation in the study. The sample size calculation was determined by the effect size, power (1-β), and significance level (α) before the recruitment. A convenience sample of community-dwelling older women was recruited from the public health center. Among the total participants, five subjects were excluded because of insufficient responses. We also chose female respondents only in this study due to distinct gender differences in the association between social isolation and depression [33] and small samples of male respondents in this study. Therefore, a sample size was calculated using the G*Power 3.1 program and had a 0.99 efficacy in detecting 0.05 effect size with two predictors at the 5% level of significance in multiple regression analysis, indicating that the number of participants in this study was sufficient. This study was reviewed and approved by the Research Ethics Review Committee of Yonsei University.

### 2.2. Measures

#### 2.2.1. Depression

Depression was measured by the Korean version of the five-item Geriatric Depression Scale (GDS-5) [34] translated by Park et al. [35]. The GDS-5 consists of five items with one “yes” and zero “no” for a total of 5 points. The five items include (1) “Are you basically satisfied with your life?”, (2) “Do you often get bored?”, (3) “Do you often feel helpless?”, (4) “Do you prefer to stay at home, rather than going out and doing new things?”, and (5) “Do you feel pretty worthless the way you are now?”. The GDS-5 total scores range from 0 to 5, and a higher score of the sum of GDS-5 items means higher depression. For internal consistency, Cronbach’s alpha for the GDS-5 was 0.76, which is at the acceptable level as shown by Hulin et al. [36].

#### 2.2.2. Loneliness

Loneliness was measured by the Korean version of DeJong Gierveld Loneliness Scale (DGLS-6) consisting of six items, which included three statements for “emotional loneliness” and the other three for “social loneliness” [37]. The “emotional loneliness” includes three negatively worded items (“I experience a general sense of emptiness”, “I miss having people around”, and “Often, I feel rejected”). The “social loneliness” includes three positively worded items (“There are plenty of people that I can lean on in case of trouble”, “There are many people that I can count on completely” and “There are enough people that I feel close to”). The items had three response categories: “no,” “more or less”, and “yes”. Consistent with DGLS-6, we scored the answer to the scale, neutral and positive answers (“more or less”, “yes”) on negatively worded items resulted in the emotional loneliness score, ranging from 0 (not emotionally lonely) to 3 (intensely emotionally lonely). In other words, “more or less”, and “yes” were scored as 1, and “no” was scored as 0. For the social loneliness score, we also counted neutral and negative answers (”no” and “more or less”) on the positively worded items resulting in the social loneliness score, ranging from 0 (not socially lonely) to 3 (intensely socially lonely). In social loneliness, “more or less” and “no” were scored as 1, and “yes” was scored as 0. The total score of DGLS-6 scales ranged from 0 to 6. A higher score indicates a higher level of loneliness. The items were translated into Korean and verified using the back-translation procedure by a translator fluent in both Korean and English. Cronbach’s alpha for the DGLS-6 was 0.77. 

#### 2.2.3. Subjective Physical Health

Subjective physical health was measured using self-rated physical health status (SPH) developed by the research team. Subjective physical health refers to how individuals evaluate their own physical health status. National Institute on Aging (NIA) and National Council on Aging (NCA) report that physical health involves many aspects of life such as: sleep well, eating well, and being physically active [38,39]. For older adults, not only eating well, but also healthy bowel and bladder function has the potential to have a marked impact on physical health [40]. In a previous study [41], a single-item of subjective health status was used as a tool to identify how individuals perceive their own health status. We designed a tool to identify how older adults perceive five dimensions (eating well, sleeping well, being physically active, healthy bowel and bladder function) of their own physical health. This questionnaire was designed to represent five basic physical health dimensions for the older adults, based on previous studies [42,43,44]. It consists of five items with one “yes” and zero “no” for a total of 5 points. The questionnaire includes (1) “Have you eaten well in the past week?” (proper eating), (2) “Have you had a bowel movement well in the past week?” (bowel movement), (3) “Have you been urinating well in the past week?” (urinary movement), (4) “Have you slept well in the past week?” (sleep), and (5) “Have you been active in the past week?” (activities). The total scores of the SPH range from 0 to 5, and a higher score of SPH means a higher perception of physical health status.

#### 2.2.4. Resilience

Resilience was measured using the Brief Resilient Coping Scale (BRCS) [45]. Resilience is conceptualized as to cope with difficult situations in a highly adaptive manner, and on an adequate scale for the older population [46]. This scale was translated into Korean language and verified using the back-translation procedure by a bilingual translator for Korean and English. The BRCS consists of four items with a 5 Likert type scale from 1 (strongly disagree) to 5 (strongly agree). The four items include (1) “I look for creative ways to alter difficult situations’’, (2) “Regardless of what happens to me, I believe I can control my reaction to it”, (3) “I believe that I can grow in positive ways by dealing with difficult situations”, and (4) “I actively look for ways to replace the losses I encounter in life”. The total scores of the BRCS range from 5 to 20, and a higher total score of BRCS means a higher level of resilience. Cronbach’s alpha for the BRCS was 0.83.

#### 2.2.5. Social Support

Social support was measured using the Modified Medical Outcomes Study Social Support Survey (mMOS-SS) [47], translated by the research team. The mMOS-SS is a tool to assess social support of the people, especially in older adults. This scale consists of eight items with a 5 Likert type scale: 1 (never), 2 (sometimes), 3 (often), 4 (mostly), and 5 (always). The eight items include “how often is someone available”: (1) to help you if you were confined to bed, (2) to take you to the doctor if you needed it, (3) to prepare your meals if you are unable to do it yourself, (4) to help with daily chores if you were sick, (5) to have a good time with, (6) to turn to for suggestions about how to deal with personal problems, (7) who understands your problems, and (8) to love and make you feel wanted. The total scores of the MOS-SS range from 5 to 40, and a higher total score indicates a higher social support. Cronbach’s alpha for the mMOS-SS was 0.94.

### 2.3. Statistical Analysis

Descriptive statistics of the study was used to identify the characteristics of the variables. Descriptive statistics was estimated as mean and standard deviation for continuous variables or percentages for categorical variables. The Pearson correlation analysis was used to examine the association among loneliness, physical health, resilience, social support, and depression. The 5000 non-parametric bootstrapping analyses were performed to test the mediational model of subjective physical health, resilience, and social support as parallel mediators of the association between loneliness and depression among older adults living alone. Bootstrap mediation does not impose the assumption of normality of the sampling distribution of the indirect effect, and therefore is considered to be more powerful for hypothesis testing for mediation analysis than the Sobel test [48,49]. 

This study used a parallel mediation model, which is a basic mediation model from Hayes PROCESS templates, to examine the mediating effects and identify which and how variables among physical (subjective physical health), cognitive (resilience), and social (social support) dimension have the mediating effect on the association between loneliness and depression. These mediators are allowed to correlate with one another, but not to influence each other in causality [50]. Potential covariates, including age, religion, and duration of living alone, were included in the analysis to examine whether they affect the results or not [50]. However, age (*b* = 0.0024, *p* = 0.87), religion (*b* = 0.0632, *p* = 0.71), and living duration (*b* = 0.0006, *p* = 0.27) were not found to be statistically associated with the level of depression on the total effects model. Thus, we did not include them in the mediation model. Data analysis was performed by using the PROCESS function V2.1. in SPSS V.21. We hypothesized that loneliness (X) would indirectly affect depression (Y) through three mediators: subjective physical health (M1), resilience (M2), and social support (M3), respectively (Figure 2).

First, the individual direct effects were calculated through the following paths:M1 on X (a1)M2 on X (a2)M3 on X (a3)Y on M1 (b1)Y on M2 (b2)Y on M3 (b3)

Y on X (c’) which is direct effect of X on Y.

Second, the specific indirect paths and total indirect effect were calculated as:Indirect effect of X on Y via M1 = a1 × b1 = a1b1
Indirect effect of X on Y via M2 = a2 × b2 = a2b2
Indirect effect of X on Y via M3 = a3 × b3 = a3b3

Total indirect effect via M1, M2 & M3 = a1b1 + a2b2 + a3b3.

Finally, the total effect (c) was found out as = a1b1 + a2b2 + a3b3 + c’.

In this model, the mediation is significant if the 95% confidence interval from the lower limit (LL) to upper limit (UL) for the indirect effect (IE) does not include zero [51]. All involved variables in the analyses were standardized (z scores) before running the analysis, hence standardized coefficients were reported for the total, direct and IEs. The completely standardized effect size (CS) was reported as indices of effect size [51]. Miocevic et al. [52] mentioned that standardized mediation effect-size measures were relatively unbiased and efficient in the parallel model. If a comparison of the two indirect effects was intended, the contrast bootstrap interval was examined. All analyses were conducted using SPSS V.21 software (SPSS Inc., Chicago, IL, USA). A *p*-value < 0.05 was considered statistically significant.

## 3. Results

### 3.1. Description of the Study Population

Among 344 participants, we excluded five participants because of insufficient responses and male respondents (*n* = 31, 9%) due to small samples in this study. As a result, this study included a total of 308 older females aged more than 60 years old living alone in a medium-sized city of South Korea. Table 1 shows the descriptive statistics of the study population. The average age of the study population was 80 years (SD = 5.89) ranging from 64 to 94 years. About 92.5% (*n* = 285) had no education or under primary school and the mean duration of living alone was approximately 13.6 years (SD = 114.39). The average values of depression and loneliness were 2.37 and 3.99, respectively. 

### 3.2. Loneliness, Subjective Physical Health, Resilience, Social Support and Depression

Table 2 shows the results of the Pearson correlation analysis. We found that depression was negatively correlated with subjective physical health (r = −0.442, *p* < 0.001), resilient coping (r = −0.368, *p* < 0.001), and social support (r = −0.310, *p* < 0.001), indicating that higher depression is associated with lower levels of the physical, psychological and social aspects. The results also revealed that loneliness was negatively correlated with subjective physical health (r = −0.295, *p* < 0.001), resilience (r = −0.345, *p* < 0.001), and social support (r = −0.315, *p* < 0.001). Loneliness had a positive correlation with depression (r = 0.428, *p* < 0.001), indicating that the more loneliness, the higher depression.

### 3.3. Results of the Parallel Mediation Model

Table 3 and Figure 3 describe the results of the parallel mediation model, including all of the direct and indirect effects. For indirect effects, the regression model predicts depression from subjective physical health, resilience, and social support. We found positive effects for subjective physical health (a1b1, *b* = 0.094), resilience (a2b2, *b* = 0.059), and social support (a3b3, *b* = 0.049). The results showed that three mediators were significantly associated with loneliness and depression. Indirect effect of loneliness on depression through subjective physical health (a1b1), holding all other mediators constant, was statistically significant, indicating that older women living alone who had low loneliness were more likely to have low depression through high perception of subjective physical health. In addition, a significantly and positively indirect effect of loneliness on depression through resilience (a2b2) means that older women living alone with low loneliness were more likely to have reduced depression through high levels of resilience. 

Lastly, we found a significantly indirect effect of loneliness on depression through social support (a3b3), indicating that older women with low loneliness were more likely to decrease depression level through high perception of social support. Therefore, all three mediators were found to significantly contribute to the overall indirect effect, controlling for all other mediators in the model. Specific indirect effect contrasts between the proposed mediators did not show statistically significant difference between the indirect effects of subjective physical health and resilience (*b* = 0.029, 95% CI: −0.032 to 0.086), subjective physical health and social support (*b* = 0.040, 95% CI: −0.010 to 0.090), and resilience and social support (*b* = −0.011, 95% CI: −0.064 to 0.040). Three positive mediations coexist in parallel, but do not differ significantly in size, as the contrast bootstrap confidence interval does not encompass zero. 

For direct effects, subjective physical health was regressed on loneliness (path a1) and the standardized coefficients reported here was −0.218, indicating that loneliness is negatively associated with subjective physical health (*p* < 0.0001). Resilience was regressed on loneliness (path a2) and the effect was significantly negative (*b* = −0.564). Social support was regressed on loneliness (path a3) and the standardized coefficient reported was −1.257, which means that loneliness was negatively predicted by social support. We found negative effects of subjective physical health (path b1, *b* = −0.429), resilience (path b2, *b* = −0.108), and social support (path b3, *b* = −0.034) on depression. All associations were statistically significant (*p* < 0.05). The direct effect (path c’) was explored by regressing depression on loneliness, which was 0.242 (*p* < 0.0001).

Table 3 and Figure 3 also show that the direct effect of loneliness on depression (c’) was statistically significant (c’ = 0.242, *p* < 0.0001). The total indirect effect via three mediators (a1b1 + a2b2 + a3b3) was significantly positive (*b* = 0.197). The results based on 5000 bootstrapped samples indicated that the total effect (a1b1 + a2b2 + a3b3 + c’) of loneliness on depression was significant (*b* = 0.439, *p* < 0.0001), indicating that those with high loneliness reported high depression even when considering the indirect effect of loneliness through all three mediators. Consequently, the total effect of loneliness on depression was due to an indirect path. The results of the parallel mediation analysis showed that loneliness was directly and indirectly related to depression through its association with the subjective physical health, resilience, and social support. Therefore, the results indicated a complementary partial mediation [53].

## 4. Discussion

This study examined the mediating effects of physical health, resilience, and social support on the association between loneliness and depression among Korea elderly women living alone. We found the partial mediating effects of physical health, resilience, and social support on the association between loneliness and depression for the elderly women living alone. This is the first study, to our knowledge, which included three factors (physical health, resilience, and social support) in the same model as the mediators of the association between loneliness and depression among the elderly living alone. The findings suggest the importance of supporting the community-based programs to improve physical, mental, and social health of the elderly people, particularly older women, as a way to minimize the level of loneliness and prevent depression.

Specifically, this study showed the strongest partial mediating effect of physical health (a1 × b1 = 0.094) on the association between loneliness and depression for the aged women who live alone. Previous studies found that older people who have feelings of loneliness were more likely to have worse physical health, including high blood pressure and low quality of sleep, and experience poorer physical and mental health than those who do not [20,21]. The other research reported that physical activity level or self-reported physical health was associated with depression in the older population [21,54]. This finding suggests that interventions to reduce depression among elderly women may require the consideration of enhancing physical health and its effects on health behaviors.

Resilience was subsequently found to have a mediating effect (a2 × b2 = 0.059) on the association between loneliness and depression for the elderly women living alone. This finding corroborates the results of earlier studies that the association between loneliness and mental health symptoms was partially mediated by resilience among the older population [24,26,27]. This also reflects that resilience could contribute to reducing negative emotions and improving well-being among older persons [25].

Among the three mediating factors, we found that social support is the weak but significant mediator (a3 × b3 = 0.043) on the association of loneliness with depression compared to other physical and mental dimensions for the older female who lives alone. Although loneliness was strongly associated with decreased social support, the magnitude of association between social support and depression is small (b3 = −0.034). Previous studies reported that the level of loneliness had a negative association with the social support from various groups of people, including children, siblings, relatives, and neighbors, in Korean elderly women [55]. However, elderly men were more vulnerable for depression than elderly women, although low social support may be related to depression in later life [56]. Given that interventions including social support may decrease depression and suicidal ideation for the older population [57], the programs for the elderly people need to have interventions to support social relationships or social networks in order to reduce the level of loneliness and depression. 

Older adults inevitably encounter life events, such as loss of partners or closeness in later life, which may increase feelings of loneliness [58]. Although people living alone do not necessarily feel lonely, elderly individuals, particularly older women, who live alone in South Korea are more likely to feel loneliness since the Asian culture has emphasized family togetherness and the interdependence of family members [59]. Such loneliness among elderly women living alone is highly associated with increased vulnerability to physical, emotional, and social health problems, related to increased levels of depressive symptoms in our study [20,21]. More specifically, we found that the average values of depression and loneliness among this study’s population were higher compared to previous studies which used the same measurements. For example, the average GDS-5 score of 139 Portuguese elderly people living in urban and rural areas was 1.04 [60], indicating that the older adults living alone in our study (score = 2.37) have higher levels of depression. Additionally, the average DLS-6 score of 1360 community-dwelling older adults in European countries was 2.16 [61], showing that the sample in this study (mean score = 3.99) was at high risk of loneliness. It is thus suggested to be targeted with tailored and multiple interventions for elderly women who live alone to prevent depressive symptoms and promote healthy aging [62].

Although loneliness can be directly associated with depression among older women living alone, we found that subjective physical health status, resilience, and social support might be crucial factors in explaining depression. We suggest that diverse and affordable community resources and multiple interventions to encourage overall physical health, promote resilient attitudes/behaviors, and improve positive relationships/social support are needed to alleviate depression in elderly women living alone. Those tailored and affordable programs for elderly women living alone are important not only for providing them with support and ability to cope with adversity, but also for encouraging health-related activities that can have a positive influence on their mental health [63]. Thus, the community programs and strategies by professional healthcare workers to improve the quality of life, well-being, and mental health should be participants-centered and inclusive for vulnerable populations, particularly elderly women living alone [64].

This study has several limitations. First, the study population of this study included the elderly women living alone recruited from only a single community center located in a middle-sized city of Korea. Additionally, we used a convenience sampling so that the sample of this study might under- or over-represent the population. Thus, the results may not be generalizable to the elderly population in other regions. Second, this was a cross-sectional observational study so that this study did not examine causal relationships. Third, the survey for physical health was developed by our research team based on several literature and did not have a pilot testing or validity evaluation. A subjective measure for this variable was used in this study so that it might cause a measurement error. Fourth, although we used a parallel mediation model in this study, alternative models might have different study findings. A future study might consider alternative models to confirm the results. Lastly, we did not include potential control variables like economic status that could affect the relationship. Therefore, future studies should consider variables, such as income level, chronic conditions, and disability [65].

## 5. Conclusions

This study showed that loneliness was directly and indirectly associated with depression through its association with the subjective physical health, resilience, and social support for the older women living alone in Korea. By using three mediators (physical health, resilience, and social support) in the same model, this study attempted to comprehensively understand the mechanisms of the association between loneliness and depression among the elderly living alone. The findings suggest the importance of enhancing physical, cognitive, and social health as an effective cognitive-behavioral treatment for depression in older people, particularly older women, as a way to minimize the level of loneliness and prevent depression. This study’s results will help healthcare professionals to consider developing interventions to enhance physical health and social support in their programs that care for the mental health of the elderly women.

## Figures and Tables

**Figure 1 ijerph-19-09246-f001:**
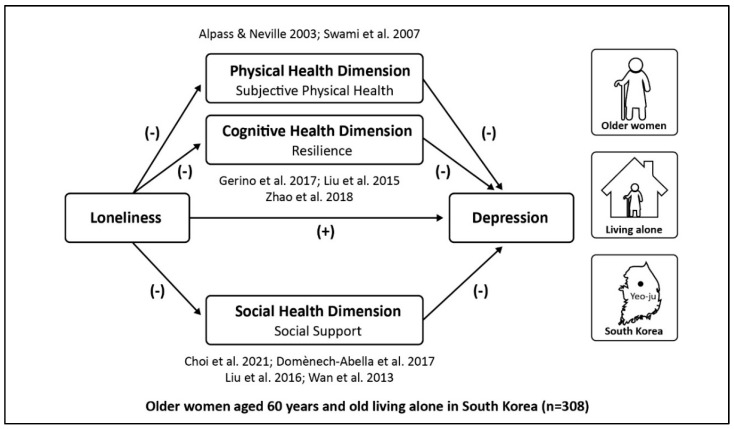
Conceptual model [11,12,21,22,24,26,27,30].

**Figure 2 ijerph-19-09246-f002:**
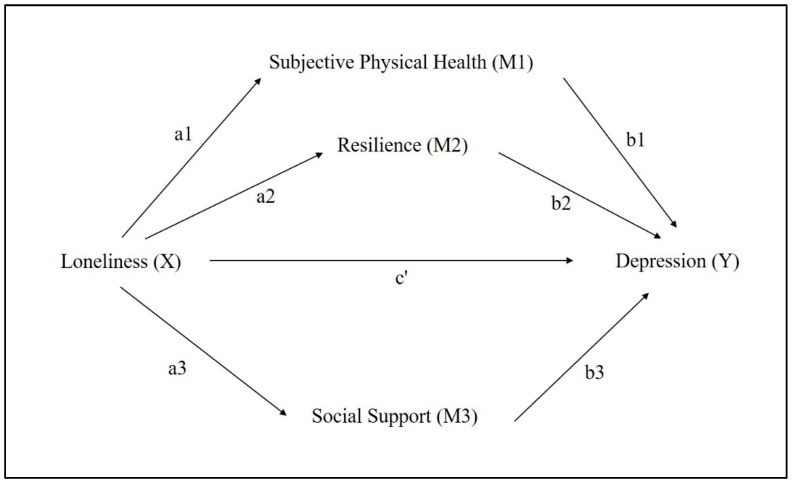
Hypothesized model.

**Figure 3 ijerph-19-09246-f003:**
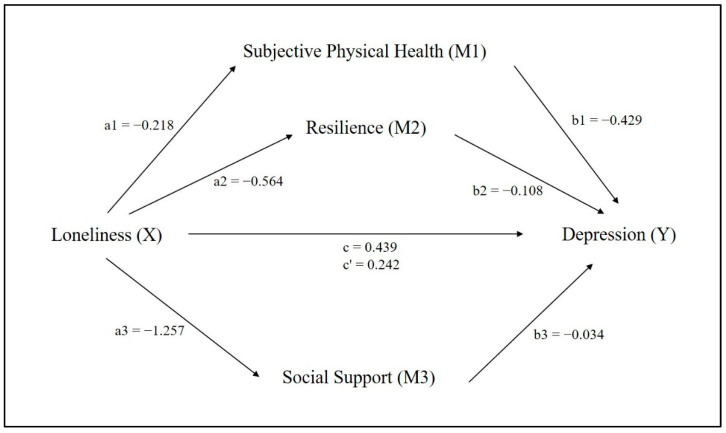
Parallel mediation model for loneliness and depression via subjective physical health, resilience, and social support.

**Table 1 ijerph-19-09246-t001:** Descriptive statistics of the study population (*n* = 308).

Variables		*n* (%) or Mean (SD/Min, Max)
Age	Years	80.02 (5.89/64, 94)
Education	None	194 (63.0%)
	Primary school	91 (29.5%)
	Middle school	16 (5.2%)
	High school over	7 (2.3%)
Religion	No	139 (45.1%)
	Yes	169 (54.9%)
Duration of living alone	Years	13.57 (114.39/0.2, 46)
Depression	Values	2.37 (1.73/0, 5)
Loneliness		3.99 (1.69/0, 6)
Subjective physical health		3.68 (1.27/0, 5)
Resilience		12.56 (2.85/5, 20)
Social support		17.00 (6.67/5, 40)

Note: SD—Standard deviation, min—minimum, max—maximum.

**Table 2 ijerph-19-09246-t002:** Results of correlation tests between the measured variables (*n* = 308).

	Loneliness	Physical Health	Resilience	Social Support	Depression
Loneliness	1	−0.295 *	−0.345 *	−0.315 *	0.428 *
Physical health	-	1	0.237 *	0.153 *	−0.442 *
Resilience	-	-	1	-	−0.368 *
Social support	-	-	-	1	−0.310 *
Depression	-	-	-	-	1

Note: * *p*-value < 0.001.

**Table 3 ijerph-19-09246-t003:** Path coefficients for parallel mediation model.

Variable/Effect	*b*	*SE*	*t*	*p*-Value	95% Confidence Interval	
					Boot-LLCI	Boot-ULCI
loneliness → physical health (a1)	−0.218	0.04	−5.31	<0.0001	−0.299	−0.137
loneliness → resilience (a2)	−0.564	0.09	−6.26	<0.0001	−0.742	−0.387
loneliness → social support (a3)	−1.257	0.21	−5.89	<0.0001	−1.676	−0.838
physical health → depression (b1)	−0.429	0.06	−6.27	<0.0001	−0.563	−0.294
resilience → depression (b2)	−0.108	0.03	−3.36	0.001	−0.171	−0.045
social support → depression (b3)	−0.034	0.01	−2.55	0.011	−0.061	−0.008
loneliness → physical health→ depression (a1b1)	0.094	0.02	-	-	0.055	0.137
loneliness → resilience → depression (a2b2)	0.059	0.02	-	-	0.023	0.101
loneliness → social support → depression (a3b3)	0.049	0.02	-	-	0.016	0.085
**Effects**						
Direct Effect (c’)	0.242	0.05	4.46	<0.0001	0.135	0.349
Total Indirect *	0.197	0.03	-	-	0.138	0.262
Total Effect (c)	0.439	0.05	8.19	<0.0001	0.334	0.545

* Based on 5000 bootstrap samples. Note: *SE*–Standard Error, LLCI–Lower Level Confidence Interval, ULCI–Upper Level Confidence Interval.

## Data Availability

The data presented in this study are available on request from the corresponding author. The data are not publicly available due to privacy issues.

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
