# Peer review of "Association between Loneliness and Depression among Community-Dwelling Older Women Living Alone in South Korea: The Mediating Effects of Subjective Physical Health, Resilience, and Social Support"

_ijerph, 2022, doi:10.3390/ijerph19159246_

Round 1

Reviewer 1 Report

Dear Authors,

I commend your effort to design, conduct, and report this study. I think this study is relevant for helping the scientific community better understand the public health issues of loneliness and depression.

Kindly see the attached file for my review comments.

Best of luck in your scientific endeavors.

Author Response

Title: Association between loneliness and depression among community-dwelling older women living alone in South Korea: The mediating effects of subjective physical health, resilience, and social support

General Comment

The topic and scope of the study is relevant for addressing the issues that affect the health and well-being of older adults. The methodological approach of this study is also commendable as the authors sought to enrich scientific understanding of the phenomenon of interest -depression- by exploring the complexities and interdependencies between antecedents of depression.

However, I think there are theoretical and methodological flaws that need to be addressed to improve the rigor of the study and the overall quality of the manuscript.

  • Response: We appreciate your valuable time and feedback. We did our best to respond to your comments. We believe that the manuscript has been improved based on your comments and suggestions.

Introduction Section

The introduction reads like an annotation of existing research on the topic. There is need to improve the introduction section to provide a theoretical basis for the selected constructs and the proposed construct relationships. Specifically, beyond simply highlighting previous studies that have explored some of the relationships between the constructs in the study, the authors should thoroughly review the literature to provide justification for why they think the constructs are related in the ways that they propose.

  • Response: Thank you for your comment. We added some information to provide a theoretical basis for the selected constructs and the proposed construct relationships and relevant references in the introduction section.

The authors should define the selected constructs with relevant citations. For instance, what exactly is meant by resilience? Are the authors conceptualizing it as a trait or an attribute that is externally imputed? What is the scope of social support being considered? An older adult may live alone but have social support in the form of community groups, friends and acquaintances. More robust construct clarification is needed.

  • Response: We appreciate your point. We have included detailed information about each selected construct in the 2.2 Measures section. We added some information to make these constructs clear.

The introduction section should also include a model or figure that shows the proposed construct relations. This will help the reader to intuitively makes sense of the content of the introduction section as there are many relationships that were discussed, and it is easy to lose track of what the main aim of the study is.

  • Response: We appreciate your suggestion. We have included a conceptual framework that can explain the proposed construct relations (Figure 1).

Lines 72-77: I suggest the authors rephrase this section. While a parallel mediation model can assess (the magnitude and direction of) mediation effects, it is not sufficient for elucidating the mechanism of construct relations.

  • Response: Thank you for your suggestions. We have rephrased this section by highlighting the research gaps in the existing literature as follows: “The previous studies have investigated the direct and indirect relationships between loneliness and depression by considering the mediating effects of physical social support, resilience, and physical health, respectively. However, there are few studies understanding these relations comprehensively. Most important, the mechanism on how and to what extent loneliness is associated with depression remains unclear among older adults, in particular older women living alone. There are also very few related studies in the context of South Korea with a rapidly aging population. Our study fills these research gaps in the existing literature by 1) exploring the direct influence of loneliness on the depression of older women aged 60 years and old living alone in South Korea and 2) exploring the mediating effects of subjective physical health, resilience, and social support, comprehensively.”

Materials and Methods

Lines 87-89: how were participants who met the inclusion criteria determined? For instance, what process was used to determine an older adult that had (or did not have) cognitive impairment? What do the authors mean by “living alone?” Please explain for each of the inclusion criteria.

  • Response: Thank you for your comments. We revised this part to explain each of the including criteria in detail as follows:

“The original inclusion criteria for the study subjects were as follows: (a) aged 60 years and older; (b) living alone at home (i.e., one person household); (c) being no diagnosed with dementia; and (d) having no problem of communication. Subjects were recruited from a public health center in the city.”

Lines 93-96: Was this a post hoc power analysis or was the power analysis done prior to recruitment? Please specify. Also, avoid reporting results under methods section. Results should go under results.

  • Response: Thank you for your point. We did the power analysis before recruitment. We added this information in the methods section. And we deleted the exact numbers of sample size in the methods section and included them in the results section as you suggested.

Measures

Line 108: Please provide citation for the cut-point for Cronbach’s alpha

  • Response: We added the citation for the cut-point for Cronbach’s alpha. Hulin, Netemeyer, & Cudeck (2001) mentioned that a general accepted rule is that alpha of 0.6-0.7 indicates an acceptable level of reliability, and 0.8 or greater a very good level. Therefore, the sentence was revised as follows: “For internal consistency, Cronbach’s alpha for the GDS-5 was 0.76, which is at the acceptable level as shown by Hulin et al (2001).”

Lines 117-122: The description of how the data from the loneliness scale were treated is unclear. Did you use different scale labels when you adapted the scale for your study? Please clarify this chunk of text.

  • Response: Thank you for your comment. We added some information to clarify this part. The ‘emotional loneliness’ includes three negatively worded items (“I experience a general sense of emptiness”, “I miss having people around”, and “Often, I feel rejected”). The ‘social loneliness’ includes three positively worded items (“There are plenty of people that I can lean on in case of trouble”, “There are many people that I can count on completely” and “There are enough people that I feel close to”). The items had three response categories: “no,” “more or less”, and “yes”. Consistent with DGLS-6, we scored the answer to the scale, neutral and positive answers (“more or less”, “yes”) on negatively worded items resulted in the emotional loneliness score, ranging from 0 (not emotionally lonely) to 3 (intensely emotionally lonely). In other words, ‘more or less’, and ‘yes’ were scored as 1, and ‘no’ was scored as 0. For the social loneliness score, we also counted neutral and negative answers (”no” and “more or less”) on the positively worded items resulting in the social loneliness score, ranging from 0 (not socially lonely) to 3 (intensely socially lonely). In social loneliness, ‘more or less, and ’no’ were scored as 1, and ‘yes’ was scored as 0.

Lines 129-130: There seems to be a word missing in the text “to represent five basic physical health of the older adults.” I suggest “to represent the five basic physical health dimensions for older adults.”

  • Response: According to your suggestion, we revised this part.

The Subjective Physical Health scale was home grown. Did you do any form of pilot testing? Did you assess for face and construct validity? Internal reliability? If so, please include these details in the report. IF not, discuss as a study limitation.

  • Response: We did not conduct pilot testing and assess and reliability for this variable. So we included this in the limitation section: “Third, the survey for physical health was developed by our research team based on several literature and did not have pilot testing or validity evaluation. A subjective measure for this variable was used in this study so that it might cause a measurement error.”

Line 154: no need to use quotations in brackets

  • Response: We agree with your comment. We deleted quotations in brackets.

Statistical Analysis

Lines 169-171: Very good to include this rationale for choice of method.

  • Response: Thank you for your comment.

Line 177: What does this mean? Including the covariates in the model statistically controls for their effects on the relationships of interest. Although they may not have had significant regression coefficients, it is best practice to report all your results. Then explain that these three covariates were excluded from further analyses. I suggest you use change in R2 as a criterion for inclusion or exclusion of predictors as significance testing is biased by sample size, while R2 gives an indication of how much variance in the outcome is explained by predictors.

  • Response: We appreciate your point. As we mentioned in the manuscript, we included potential covariates, including age, religion, and duration of living alone, in the analysis to examine whether they affect the results or not [39]. However, Age (b=.0024, p=.87), religion (b=.0632, p=.71), and living duration (b=.0006, p=.27) were not found to be statistically associated with the level of depression on the total effects model. Thus, we did not include them in the mediation model. We revised this part.

Please include a brief justification for your choice of using a parallel mediation model over other alternatives like a serial mediation model or even a moderation analysis and discuss the underlying assumptions for your model choice. For instance, an assumption of the parallel mediation model is that there are no interactions between the mediators. Whereas, it can be argued conceptually that social support is a moderator of the relationship between loneliness and depression. I suggest that the authors search the literature and ground their model selection in theory.

  • Response: Thank you for your comment. We used a parallel mediation model, which is a basic mediation model from Hayes PROCESS templates, to examine the mediating effects and identify which and how variables among physical (subjective physical health), cognitive (resilience), and social (social support) dimension have the mediating effect on the association between loneliness and depression. These mediators are allowed to correlate with one another, but not to influence each other in causality (Hayes, 2013). A previous study (Liu, Gou, & Zuo, 2016) mentioned the effect of loneliness on depression and the mediating effect of social support.

Line 200-202: Please include a rationale for standardization. Why was it important to standardize the variables? How was it done? Your readers should have as much information as possible to make sense of the study without having to refer to some obscure citation(s).

  • Response: Thank you for your comment. We standardized the variables given that Miocevic, O’Rourke, Mackinnon, & Brown (2018) mentioned that standardized mediation effect-size measures were relatively unbiased and efficient in the parallel model. We included this information in the Methods section (2.3. Statistical analysis).

Line 205: “statistically” is missing before “significant”

  • Response: We added “statistically” in this part. “A p-value <0.05 was considered statistically significant.”

Results

Tables need to be redesigned and formatted to make them more readable.

  • Response: We appreciate your point. We combined Tables 1 and 2 to a table (Table 1) to make them more readable.

Line 213: Please state “major” variables more specifically

  • Response: We added the information in Table 2 to Table 1 and revised this part, deleting a word of “major”.

In mediation models, the path for X to Y, independent of M, is what is referred to as direct effect. I suggest you do not refer to the path from X to M as direct effect to avoid confusion. Moreover, the interpretation of the regression coefficients for the paths X to M is typically not quite informative.

To better and more insightfully organize your results section, I suggest you present your results for:

The three specific indirect effects (X through M to Y)

The total indirect effect (sum of all the paths from X through M to Y)

The direct effect (X to Y independent of M), and

Total effect (total indirect effect plus direct effect)

  • Response: Thank you for your suggestion. We reorganized the results section to make it clear based on your suggestion.

Line 254: I suggest you say “we found positive mediator effects”

  • Response: We appreciate your suggestion. We revised this part, adding “positively” in the previous sentence as follows: “Lastly, we found a significantly and positively indirect effect of loneliness on depression through social support (a3b3), indicating that older women with low loneliness were more likely to decrease depression level through high perception of social support”

The interpretation of indirect effects should include the caveat: “controlling for all other mediators in the model”

  • Response: We agree with you comment. We added this to the manuscript.

Do contrasts to determine whether specific indirect effects are different from each other. You can easily specify this test in the SPSS PROCESS procedure. This test will provide empirical support for (or against) your theoretical proposition that these mediator effects are distinct from one another, and also allow you to make more contribution to literature. Refer to this material (Hayes, A. F. (2013). Mediation, moderation, and conditional process analysis. Introduction to mediation, moderation, and conditional process analysis: A regression-based approach edn. New York: Guilford Publications, 1, 20.)

  • Response: Thank you for your point. We included the reference (Hayes, 2013) according to your suggestion. And we mentioned that specific indirect effect contrasts between the proposed mediators did not show a statistically significant difference between the indirect effects of subjective physical health and resilience (b=.0287, 95% CI: LL=-0.032 to 0.086), subjective physical health and social support (b=.0397, 95% CI: LL=-0.010 to 0.090), and resilience and social support (b=-.0110, 95% CI: LL=-0.064 to 0.040). Three positive mediations coexist in parallel, but do not differ significantly in size, as the contrast bootstrap confidence interval does not encompass zero.”

Discussion

The discussion needs to be improved. All the constructs- antecedent, mediators, and consequent- have been well researched in literature. Rather than merely listing other similar studies, the authors should thoroughly search the literature and show how their study findings connect to and add to the literature on depression and loneliness. More specifically, the authors need to demonstrate how their study has (or has not) filled gaps in the current literature. Parallel mediation model is an empirical way to explore complex construct relations. The authors should discuss how their study has helped the scientific community to better understand the construct relations studied. One approach is to discuss the difference between the total indirect effect and the direct effect in explaining the total effect. That is, if the total indirect effect was larger than the direct effect, that suggest that mediation plays a role in explaining the relationship between loneliness and depression. Use these empirically determined differences in paths to drive your discussion.

  • Response: Thank you for your comments. We revised the discussion section accordingly as you suggested. We have connected our results/findings based on the literature in responding to how this study has filled gaps in the current literature.

In discussing Figure 2, highlight the importance of the signs of the regression coefficient, and how the signs reverse, for understanding the construct relations assessed in this study. Also, discuss implications of these construct relations for how interventions may be designed to address depression in older women living alone. All discussion points should be grounded in literature.

  • Response: We appreciate your point. We revised the discussion section based on your comments. We included the information about the construct relations and their implications with the literature.

Lines 315 – 318: Avoid the use of causal language (“leading to…”). Include citation(s) for this chunk of text.

  • Response: Thank you for your suggestion. We changed this language to “related to” and included some references for this information.

Lines 323 – 326: It appears that there is an omission or typographical error in this chunk of text. It’s incomplete and hard to understand.

  • Response: Thank you for your correction. We revised this part to complete the sentence as follows: “We suggest that diverse and affordable community resources and multiple interventions to encourage overall physical health, promote resilient attitudes/behaviors, and improve positive relationships/social support are needed to alleviate depression in elderly women living alone.”

Need to discuss study limitations

  • Cross-sectional study
  • Convenience sampling
  • Concerns about generalizability
  • The implications of alternative models for the study findings

Etc.

  • Response: We appreciate your suggestion. We revised the limitation section, adding the limitations of convenience sampling and the implications of alternative models for the study findings. We included limitations regarding cross-sectional study and concerns about generalizability in the original manuscript.

Reviewer 2 Report

Comments and suggestions for authors

I read with interest the manuscript entitled ‘Association between loneliness and depression among community-dwelling older women living alone in South Korea: The mediating effects of subjective physical health, resilience, and social support’.

The authors investigated the association between loneliness and depression among the elderly female population living alone and identified that subjective physical health, resilience and social support have a mediating effect on this association.

The manuscript is clearly presented and organized.

Materials and methods

Regarding the inclusion criteria, the authors should specify:

- the reason why they chose only women for their study, considering the fact that although among the elderly population in the mentioned region the rate of women is higher than of men, the difference (53% vs 47% (line 86) is not so significant.

- how absence of cognitive impairment was assessed

 Regarding the study population, the authors calculated the sample size (n=308) (line 94) and used a convenience sampling (line 82-83). The authors should specify more clearly which was the answer rate (how many participants were initially interviewed, how many were excluded).

The results are presented in an organized manner.

Still, is not clear if, in the case of loneliness – subjective physical health, the effect is positive (line 236) or negative (table 4 and figure 2). The authors should clarify this aspect.

The authors should also clarify what ‘subjective physical health’ means.

The discussions are well presented.

The authors should develop the conclusions section with a clearer presentation of the usefulness of the present research.

The authors should improve the references; out of a total of 53 references, 29 are prior to 2017.

Other observations

Table 4 (page 4) – the rows should be arranged (the numbers corresponding to a1b1, a2b2, a3b3)

Line 290-292 – the statement should be reformulated (it is probably about high blood pressure and low quality of sleep).

08.07.2022

Author Response

Comments and suggestions for authors

I read with interest the manuscript entitled ‘Association between loneliness and depression among community-dwelling older women living alone in South Korea: The mediating effects of subjective physical health, resilience, and social support’.

The authors investigated the association between loneliness and depression among the elderly female population living alone and identified that subjective physical health, resilience, and social support have a mediating effect on this association.

The manuscript is clearly presented and organized.

  • Response: We appreciate your review and comments.

Materials and methods

Regarding the inclusion criteria, the authors should specify:

- the reason why they chose only women for their study, considering the fact that although among the elderly population in the mentioned region the rate of women is higher than that of men, the difference (53% vs 47% (line 86) is not so significant.

- how absence of cognitive impairment was assessed

  • Response: Thank you for your comments. We included only older women in this study since most of the older adults living alone in the city were female (about 71.4%), there are distinct gender differences in the association between social isolation and depression [33], and this study includes small samples of male respondents (n=31, 9%). Regarding cognitive impairment, we included elderly women who were not diagnosed with dementia. The public health center had information about a dementia diagnosis. We changed this part to make it clear.

Regarding the study population, the authors calculated the sample size (n=308) (line 94) and used convenience sampling (line 82-83). The authors should specify more clearly which was the answer rate (how many participants were initially interviewed, how many were excluded).

  • Response: Thank you for your point. We included the information about the selection of participants as follows: “A convenience sample of 344 community-dwelling older women was recruited from the public health center. Among 344 participants, five subjects were excluded because of insufficient responses. We also chose female respondents only in this study due to distinct gender differences in the association between social isolation and depression [33] and small samples of male respondents (n=31, 9%) in this study.”

The results are presented in an organized manner.

Still, is not clear if, in the case of loneliness – subjective physical health, the effect is positive (line 236) or negative (table 4 and figure 2). The authors should clarify this aspect.        

  • Response: We confirmed that the longlines and subjective physical health had negative association.

The authors should also clarify what ‘subjective physical health’ means.

  • Response: Thank you for your point. We added some information to clarify the meaning of ‘subjective physical health’ to the Measures section.

The discussions are well presented.

The authors should develop the conclusions section with a clearer presentation of the usefulness of the present research.

  • Response: Thank you for your comment. We added some information about the usefulness of the present study in the conclusions section.

The authors should improve the references; out of a total of 53 references, 29 are prior to 2017.

Response: We appreciate your comment. We did our best to add the recent references.

Other observations

Table 4 (page 4) – the rows should be arranged (the numbers corresponding to a1b1, a2b2, a3b3)

  • Response: We revised the rows in Table 4 (Table 3 in the revised version) according to your comment.

Line 290-292 – the statement should be reformulated (it is probably about high blood pressure and low quality of sleep).

  • Response: Thank you for your point. We revised this part to “high blood pressure and poor sleep quality.”

08.07.2022

Reviewer 3 Report

Dear authors

Dear Authors

Your manuscript is very interesting, well structured, and clearly presents the process of development, analysis, and conclusions.

Congratulations.

Author Response

Thank you for your time and review. We really appreciate your comments.